# An Evaluation of Four Supraglottic Airway Devices by Paramedics in a Simulated Condition of Entrapped Trauma Patients—A Randomised, Controlled Manikin Trial

**DOI:** 10.3390/healthcare13121404

**Published:** 2025-06-12

**Authors:** Dawid Aleksandrowicz, Paweł Mickowski, Mariusz Gawrysiak, Paweł Ratajczyk

**Affiliations:** 1Department of Anaesthetics, Intensive Care Medicine and Pain Therapy, Mazovian Specialist Hospital, 26-600 Radom, Poland; 2Tytus Chałubiński District General Hospital, 34-500 Zakopane, Poland; micek123@gmail.com; 3Department of Medicine and Healthcare Sciences, University of Applied Sciences, 34-400 Nowy Targ, Poland; 4Department of Anesthesiology and Intensive Therapy, Medical University of Lodz, 90-153 Lodz, Poland; mariuszanest@gmail.com (M.G.); pawel.ratajczyk@umed.lodz.pl (P.R.)

**Keywords:** supraglottic airway device, road traffic accident, cervical spine immobilisation, entrapped patients, manikin study

## Abstract

**Introduction:** Supraglottic airway devices play an important role in airway management in both pre-hospital as well as in-hospital settings. They are a well-recognised alternative to definitive airways in current medical practice. However, despite their wide use in clinical practice, little is known about their performance in patients with restricted access. This study aims to evaluate the time required to insert a supraglottic airway device and achieve a successful ventilation of four different devices in a simulated condition of an entrapped trauma patient with simultaneous cervical spine immobilisation. The ease-of-use and first-attempt success rate were also assessed. **Methods**: Fully qualified paramedics participated in this randomised, controlled manikin trial. A manikin with the cervical collar on was placed on the driver’s seat of a passenger car. Access to the manikin was only allowed from the front. The I-gel, the SLIPA, the LMA Supreme, and the Ambu AuraGain were evaluated. The time required to insert the device and achieve successful ventilation was recorded. The first-attempt success rate and the ease-of-use by the operator were also assessed. **Results**: The LMA Supreme required the shortest mean time to insert and ventilate the manikin, 10.5 s (±1.7) vs. 16.4 s (±8.4), *p* < 0.001. The use of the LMA Supreme was associated with the highest first-attempt success rate—88%. The SLIPA device outperformed all other studied devices with regard to ease-of-use and user-friendliness. Its mean score was 8.3 out of 10. **Conclusions**: The LMA Supreme was superior in terms of both the insertion-to-ventilation time as well as the first-attempt success rate. The SLIPA device was found to be easier to use and more user-friendly.

## 1. Introduction

The introduction of supraglottic airway devices (SADs) into clinical practice in the mid-1980s revolutionised the previous approach to airway management worldwide [1,2]. SADs have become an important part of various difficult airway algorithms in both pre-hospital as well as in-hospital settings. SADs are a well-recognised alternative to definitive airways in current medical practice, making them very useful tools during the management of both routine and difficult airways [3,4,5].

The original design of a single-lumen classic laryngeal mask airway (LMA) has evolved into different modifications and further developments, leading to the creation of new double-lumen SADs. Furthermore, there is also another specific group of SADs. It aims to facilitate tracheal intubation. Such a feature may be particularly important during airway management of the injured.

Road traffic accidents (RTAs) present a significant challenge to healthcare systems in the modern-day world. According to the latest data provided by the World Health Organization (WHO), RTAs are the leading cause of death in those aged between 5 and 29 years [6]. Furthermore, road traffic collisions also form the main mechanism of injury in major trauma. This large number of victims is a burden on the economy, even in well-developed countries. Therefore, it is of utmost importance to try to reduce the number of deaths by implementing the latest guidelines and algorithms for the management of trauma patients [7,8]. In this process, an important role is played by airway management performed promptly by a skilful practitioner.

Airway management may be very difficult to perform in RTA victims, particularly in those who are unable to leave the vehicle unassisted [9]. Such patients are considered entrapped. Although definitive airway, e.g., intubation, is considered the gold standard in such circumstances, it may be difficult or impossible to perform [10,11]. In such settings, SADs may be a good alternative to definitive airways not solely for the maintenance of airway patency but also for facilitating intubation [12]. While conventional laryngoscopes as well as video and optical laryngoscopes have been extensively investigated in various settings [12,13,14,15,16,17], the data on alternative airways such as SADs in entrapped trauma patients are relatively sparse. As mentioned above, supraglottic airways may be first-choice tools for airway management of the injured, and it is important that they are thoroughly evaluated in order to recommend the equipment that performs best in this challenging scenario.

There are various types of SADs available to use. However, a pre-hospital care provider should be aware of those that perform best and, therefore, which one to use for airway management of entrapped RTA victims. Careful evaluation and assessment of currently available SADs in trauma-related scenarios play an important role in identifying the best tool. Adequate airway equipment together with experienced personnel is crucial in reducing mortality and improving patients’ outcomes. This can be achieved by confident airway management performed promptly.

Simulation-based training (SBT) has revolutionised the approach to specialty training not only in medicine but also in other areas of healthcare-related education. SBT is crucial in paramedic training [18]. It allows them to practice and develop skills, improve decision-making, and build confidence in a safe and controlled environment before facing real-life emergencies [19]. This is of particular importance concerning airway management. In Poland, airway management-related SBT is mandatory not only among healthcare professionals, e.g., paramedics and doctors, but also among members of other services such as the police and fire services. The LMA Classic and derivatives, as well as the I-gel, are the most frequently used SADs for training. Of note is the fact that simulation was used as a research method in this study.

This study aimed to evaluate four SADs, the LMA Supreme (Teleflex Inc., Wayne, PA, USA), the I-gel (Intersurgical Ltd., Wokingham, UK), the AuraGain (Ambu A/S, Ballerup, Denmark), and the SLIPA (SLIPA Medical Ltd., London, UK), for ventilation in a simulated condition of an entrapped trauma patient with simultaneous cervical spine immobilisation. The time required to insert a supraglottic airway device and achieve a successful ventilation was recorded, and the ease-of-use and first-attempt success rate were also assessed. All airway devices were evaluated by experienced paramedics in a situation when access to the RTA victim was difficult.

## 2. Methods

### 2.1. Ethical Consideration

This study has been approved by the University of Radom Ethics Committee (KB/11/2023, head: Prof. Z. Stojcev, date: 11 October 2023). Trial registration ClinicalTrials.gov: NCT06545903 (Study Registration Dates: 16 July 2024). A written informed consent was obtained from all study participants.

### 2.2. Sample and Data Collection

Fully qualified and active paramedics participated in this study. They all volunteered to participate and had previous experience that varied between four and eight years after completion of their training. Information about the study was posted on social media, and paramedics could express their willingness to take part in the study via the Internet. The area of recruitment of the study participants was limited for practical and logistical reasons to southern Poland. Figure 1 shows the study flow diagram. All paramedics who participated in this study had never used the SLIPA and the AuraGain devices.

### 2.3. Study Design and Setting

A 10 min lecture was delivered before the start of the study. It explained how to use the evaluated devices. After the theoretical introduction, all participants could familiarise themselves with the supraglottic airway devices and practice for 30 min. A skill station was set up. It consisted of the following: an AT Kelly Torso intubation manikin (Laerdal Medical AS, Stavanger, Norway) with an applied Patriot^®^ cervical collar (Össur hf., Reykjavik, Iceland) and a manual resuscitator (Ambu A/S, Ballerup, Denmark) that was used for ventilation and readily available to the study participants. Following the initial practice, the intubation manikin with the cervical collar on was placed on and secured to the driver’s seat of a FIAT Bravo passenger car (FIAT S.p.A., Turin, Italy). The car was then positioned on its left side. Firemen from a local fire brigade secured the car in place (Figure 2). An opening was created after removal of the windscreen, and access to the manikin was only allowed through it, i.e., from the front. The entire study, including data collection, took place in July 2024 on the grounds of a local ambulance station.

### 2.4. Study Procedures and Instruments

A single-digit number was allocated to each of the four studied devices, i.e., 1 for the SLIPA device (Figure 3), 2 for the I-gel (Figure 4), 3 for the LMA Supreme (Figure 5), and 4 for the Ambu AuraGain (Figure 6). A single sheet of paper with the number printed on it was placed in a dark, opaque envelope. The SPSS 29.0 software (IBM Corp., Armonk, NY, USA) was used for random allocation of these sheets. Each study participant was asked to randomly pick up an envelope and was then given the corresponding airway device to use. The maximum number of insertion attempts was limited to three per device. The time required to insert the device and achieve a successful ventilation (T_iv_) was recorded. It was measured using a stopwatch on a mobile phone (Apple Inc., Cupertino, CA, USA). The T_iv_ was measured from the moment when a device was picked up by a study participant until the ventilation was confirmed. All measurements from start to finish were taken by an independent observer. The efficacy of the insertion/ventilation and the ease-of-use by the operator were also assessed. They were measured using a numerical rating scale (NRS). On this 11-point scale, 0 corresponded to a very difficult-to-use device and 10 indicated an easy-to-use device. All paramedics used each of the four supraglottic airway devices. A failed insertion and ventilation were defined as an insertion attempt that lasted longer than two minutes or an attempt during which the manikin could not be ventilated. Another attempt was only allowed to those who failed to ventilate or insert an SAD within 2 min.

### 2.5. Data Analysis

All the collected data were analysed using Microsoft Office Excel 2021 (Microsoft Corporation, Redmond, WA, USA) and Statistica 14.0 (TIBCO Software, Inc., Palo Alto, CA, USA). The Kolmogorov–Smirnov test was used to determine whether the analysed variables matched the characteristics of a normal distribution. The Wilcoxon signed-rank test and paired Student *t*-test were used for data analysis. Assuming that the overall success rate of intubation in patients with a difficult airway would be 90% (α = 0.05, 2-sided, β = 0.1, 95% CI, *t*-value ± 2.77), the calculated sample size comprised 45 participants. A drop-out rate of about 10% was allowed, and therefore the final adjusted sample size was 50, and this was the final number of participants enrolled in the study. A *p*-value of less than 0.05 (*p* < 0.05) was considered statistically significant. Cohen’s d was used for effect size calculations.

## 3. Results

Fifty fully qualified and experienced paramedics participated in this study. Their experience ranged between four and eight years of active pre-hospital work after finishing emergency medicine training (mean 6.4 years). The majority of the participants were male (*n* = 33) compared to female (*n* = 17).

The LMA Supreme required the shortest mean time to insert and ventilate the manikin, 10.5 s (±1.7) vs. 16.4 s (±8.4) for the SLIPA, *p* < 0.001 (Table 1). The longest insertion-to-ventilation mean time was achieved when the SLIPA was used.

The SLIPA device was the only SAD under study that had not achieved a 100% insertion success rate. In this case, the overall failure rate was 4% (Table 2). The LMA Supreme had the highest first-attempt success rate—88%. A similar result was achieved when the I-gel was used (82%).

The SLIPA device outperformed all other SADs with regard to ease-of-use and user-friendliness. Its mean score was 8.3 out of 10, *p* < 0.001 (Table 3). The remaining SADs achieved significantly lower NRS scores, with the I-gel found to be the least easy to use.

## 4. Discussion

Airway management in out-of-hospital settings is often difficult and challenging, particularly in trauma patients. Those of them unable to leave the vehicle are considered entrapped. Such patients form a specific population of RTA victims, as there is often difficult or limited access to them [20,21]. This renders airway management and the entire resuscitation process extremely difficult [9].

Supraglottic airway devices have been widely used both in pre-hospital and in-hospital settings since their inception and introduction to clinical practice in the second half of the 1980s [22,23]. SADs play an important role in various difficult airway guidelines in trauma patients worldwide [24,25]. They are considered an alternative to tracheal intubation (TI) and may become very useful in certain situations [26]. In some countries, especially those with a paramedic-led service, there is a visible trend to shift from TI towards SAD use during airway management, particularly in the pre-hospital setting.

This study aimed to evaluate four different supraglottic airway devices, the LMA Supreme, the Ambu AuraGain, the SLIPA, and the I-gel, for airway management with simultaneous cervical spine immobilisation in a simulated condition of an entrapped trauma patient. All devices under study were evaluated by fully qualified paramedics in a scenario of difficult access to an RTA victim. To the authors’ knowledge, this is the first study that compared various supraglottic airway devices in a simulated condition of an entrapped RTA victim. Of note is the fact that the number of cars is on the increase worldwide [27,28]. This may lead to an increased number of RTAs, which in turn may directly pose a higher risk of road traffic accident victims being entrapped in their vehicles.

The evidence of SADs performance within the trauma patient population, especially entrapped road traffic accident victims, is sparse in the current literature.

The insertion and ventilation time (T_iv_) formed the primary outcome measure of the study. The LMA Supreme required the shortest mean time to insert and successfully ventilate the manikin, with the mean T_iv_ being 10.5 s (±1.7). This is of particular importance as airway management in trauma patients with restricted access should be performed rapidly and efficiently. In a recent study by Pap et al., four airway management devices were compared when utilised by paramedics in a simulated entrapped patient [29]. However, the studied devices were heterogeneous, and they comprised both SADs and laryngoscopes. The authors found the LMA Supreme to be superior with regard to the mean first successful ventilation time. This finding was similar to the results of our study, although Pap and colleagues achieved longer insertion and successful ventilation times than those in our study, 16.7 s vs. 10.5 s.

Use of the LMA Supreme was associated with the highest first-attempt success rate, at 88%. This device outperformed the remaining SADs, with the I-gel also being successful, achieving a first-attempt success rate of 82%. In the study by Pap and colleagues, the LMA Supreme had a 100% first-attempt success rate. In our study, the user-friendliness or the ease-of-use was also evaluated. All the devices under study were rated using the eleven-point NRS. The score ‘10’ described the easiest-to-use tool. The SLIPA device was found to be the most user-friendly of the studied SADs. It achieved a mean score of 8.3 out of 10, *p* < 0.001. The I-gel was the least user-friendly. In the abovementioned study by Pap and colleagues, the degree of difficulty and clinician preference were also assessed. Of the evaluated SADs, the LMA Supreme was rated as the least difficult to use and the preferred airway management device. Similar results were noted in our study, although the SLIPA device narrowly outperformed the LMA Supreme, 8.3 (0.9) vs. 7.8 (3.7).

With a limited number of studies available on SAD use for airway management in entrapped patients, it is very difficult to compare the results of our study. The vast majority of studies evaluated both SADs and other airway devices, mainly standard or video laryngoscopes [30].

Martin et al. evaluated the King LT-D (Laryngeal Tube-Disposable) supraglottic airway and compared it with direct laryngoscopy, digital intubation, and video laryngoscopy [31]. They assessed the one-pass success rate as well as the one-pass median placement time. The King LT-D offered significantly faster airway placement with a significantly higher first-pass success rate. Although the King LT-D and the LMA Supreme are two different SAD designs, the results showed that their performance with the entrapped patient population is superior to other airway devices. This makes them a valuable alternative concerning the airway management of trauma patients with difficult or limited access. In another study, three SADs were evaluated in a simulated condition of an entrapped patient who was difficult to access [32]. Their assessment was against the standard Macintosh laryngoscope blade, which was used as a reference. The use of the I-gel was associated with the shortest mean insertion-to-ventilation time, 11.5 s ± 6.9. This finding was contradictory to the results of our study, where the I-gel was the third fastest with regard to the mean successful insertion and ventilation time of 14.2 s ± 5.6. This was rather unanticipated as the I-gel is one of the SADs without an inflatable cuff and theoretically should be associated with shorter insertion and ventilation times. One of the reasons for this may be attributed to the ‘bulky’ design of the I-gel.

Data on the performance of both the Ambu AuraGain and the SLIPA are also very limited in the current literature. No studies exist with regard to the use of the SLIPA device within the trauma patient population. However, the SLIPA was evaluated by novice military operators [33]. In this study, five supraglottic airway devices were assessed. The SLIPA device was significantly outperformed by the other four SADs, i.e., the LMA Supreme, the LMA ProSeal, the I-gel, and the LT-D, with regard to the first-attempt success rate, insertion time, and oropharyngeal seal pressures. Similar observations were reported in our study, where the LMA Supreme was superior in terms of most study parameters. Furthermore, the SLIPA was found to be easy or very easy to use in a clinical setting [34]. Similar findings were observed in our study. However, one must bear in mind that an easy-to-use device does not always perform better in terms of insertion and ventilation times or efficacy. The Ambu AuraGain was also reported to be an easy-to-use and effective tool for airway maintenance of entrapped patients when used by firemen [35].

Despite a significantly small number of studies regarding SADs use in entrapped patients, there may be an improved way of confirming the correct placement of supraglottic airway devices, thus increasing safety, as there has been an increase in the development of new devices and techniques [36]. They may become useful in the trauma patient population in the foreseeable future. One of such new devices is the SaCoVLM, which combines a supraglottic airway device with a video system that includes a separate screen [37]. Furthermore, new vision-guided SADs insertion techniques have also been recently described [38]. Such an approach may improve the quality of airway management, thus increasing patients’ safety both in-hospital as well as in pre-hospital settings [39].

Several limitations exist concerning our study. One of them is that it was carried out with an airway manikin. It may be more difficult to insert supraglottic airway devices in real patients. On top of this, the presence of saliva, vomitus, or blood may complicate the use of SADs. However, of note is the fact that manikin studies form an important stage in a new airway device assessment process before clinical evaluation [40]. Another limitation is a relatively small sample size that comprised fifty participants. Furthermore, one has to be aware that the inclusion of experienced paramedics may be a potential source of selection bias and that results might vary when devices are used by less experienced healthcare professionals. Lack of previous experience with some of the studied devices can potentially be a source of bias and is another limitation of our study.

Presented in part at the 21st International Congress of the Polish Society of Anaesthesiology and Intensive Therapy, 12–14 September 2024, Gdańsk, Poland.

## 5. Conclusions

This study has found that the LMA Supreme was superior in terms of both the insertion-to-ventilation time as well as the first-attempt success rate. The SLIPA device was found to be easier to use and more user-friendly. However, the overall performance of the evaluated devices identified the LMA Supreme as the SAD of choice for airway management in an entrapped trauma patient with restricted access.

## Figures and Tables

**Figure 1 healthcare-13-01404-f001:**
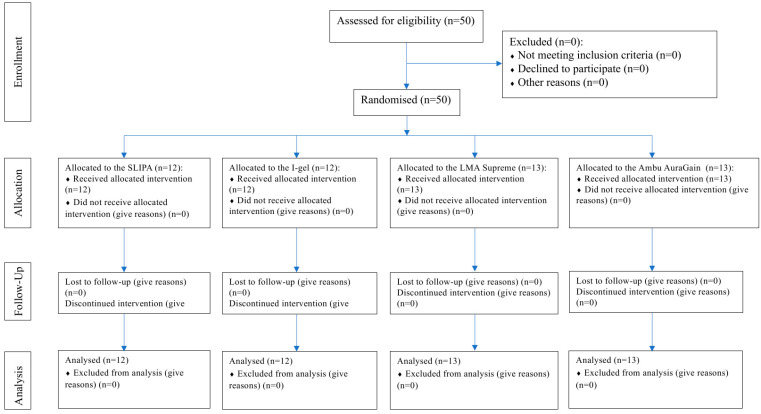
Study flow diagram.

**Figure 2 healthcare-13-01404-f002:**
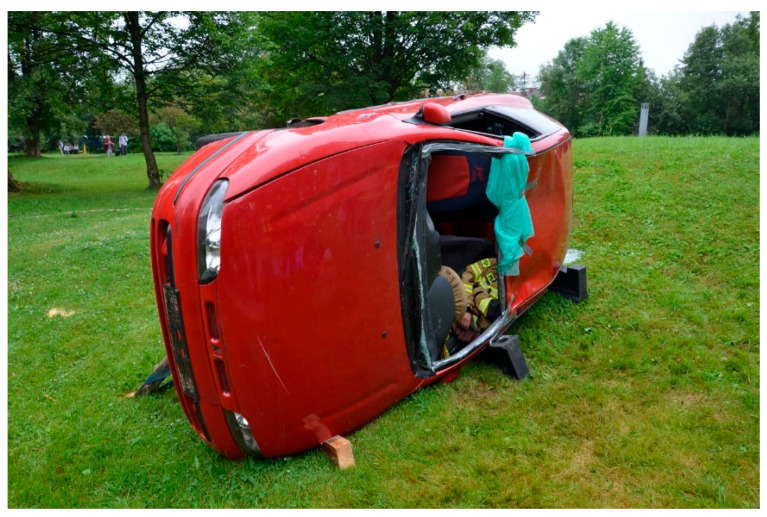
Final position of the passenger car used for the study.

**Figure 3 healthcare-13-01404-f003:**
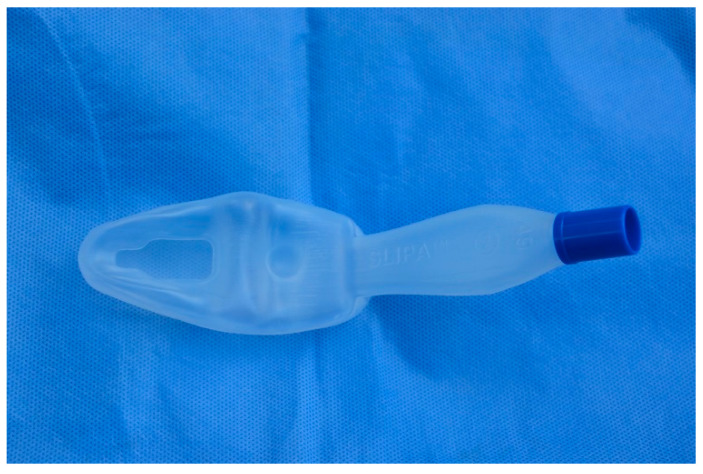
The SLIPA device.

**Figure 4 healthcare-13-01404-f004:**
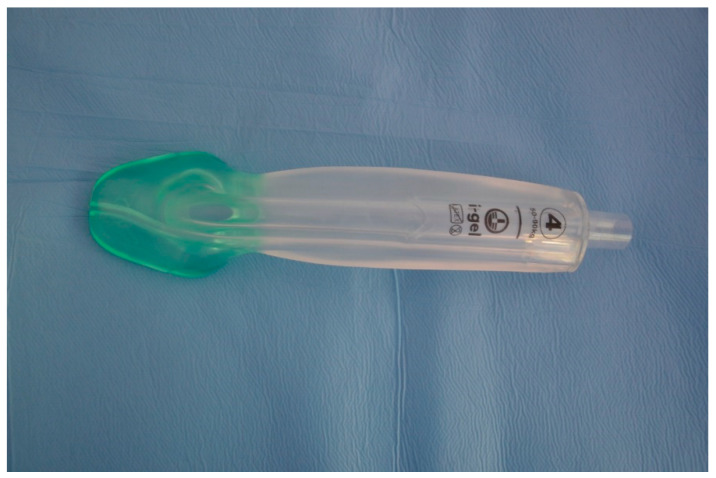
The I-gel.

**Figure 5 healthcare-13-01404-f005:**
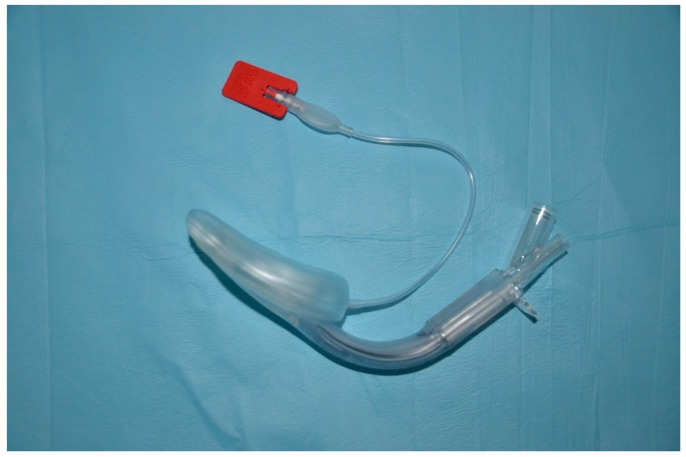
The LMA Supreme.

**Figure 6 healthcare-13-01404-f006:**
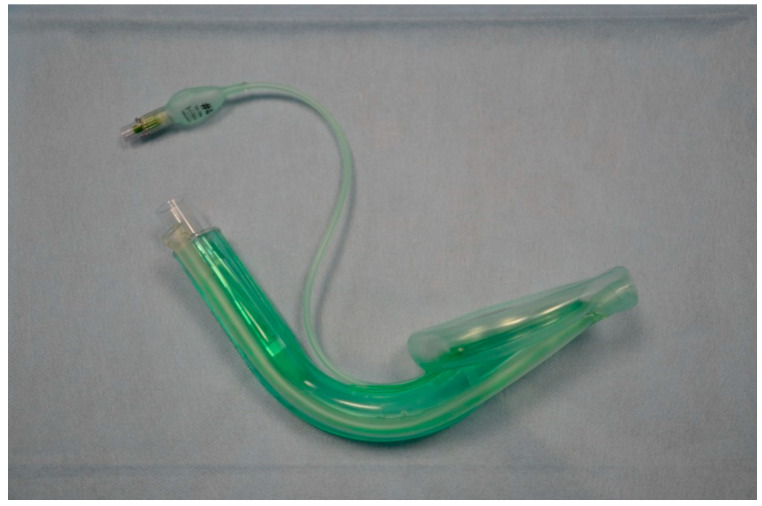
The Ambu AuraGain.

**Table 1 healthcare-13-01404-t001:** Insertion and ventilation time (T_iv_).

Airway Devices	Time Required to Insert the Device and Ventilate the Manikin (T_iv_) [s]				
	Min	Max	Mean (SD)	*p*-Value	Cohen’s d
LMA Supreme	6.8	18.8	10.5 (1.7)	0.003	-
Ambu AuraGain	7.6	20.1	12.7 (4.6)	0.17	−0.63
SLIPA	9.8	35.3	16.4 (8.4)	<0.001	−0.97
I-gel	7.5	24.5	14.2 (5.6)	0.04	−0.89

SD—Standard Deviation, s—Seconds, LMA—Laryngeal Mask Airway, SLIPA—Streamlined Liner of the Pharynx Airway.

**Table 2 healthcare-13-01404-t002:** Efficacy of the studied devices.

Airway Devices	Number of Attempts Required for Successful Placement [*n* (%)]		
	1st Attempt	2nd Attempt	3rd Attempt
LMA Supreme	44 (88)	6 (12)	-
Ambu AuraGain	39 (78)	11 (22)	-
SLIPA	35 (70)	10 (20)	5 (10)
I-gel	41 (82)	9 (18)	-

LMA—Laryngeal Mask airway, SLIPA—Streamlined Liner of the Pharynx Airway.

**Table 3 healthcare-13-01404-t003:** User-friendliness of the studied devices.

Airway Devices	NRS				
	Min	Max	Mean (SD)	*p*-Value	Cohen’s d
LMA Supreme	6	10	7.8 (3.7)	0.02	-
Ambu AuraGain	6	9	7.7 (4.1)	0.007	0.02
SLIPA	7	10	8.3 (0.9)	<0.001	−0.19
I-gel	6	9	7.5 (2.3)	0.009	0.09

NRS—numerical rating scale, SD—standard deviation, LMA—laryngeal mask airway, SLIPA—Streamlined Liner of the Pharynx Airway.

## Data Availability

The original contributions presented in this study are included in the article. Further inquiries can be directed to the corresponding author.

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
