# Peer review of "An Evaluation of Four Supraglottic Airway Devices by Paramedics in a Simulated Condition of Entrapped Trauma Patients—A Randomised, Controlled Manikin Trial"

_healthcare, 2025, doi:10.3390/healthcare13121404_

Round 1

Reviewer 1 Report

Comments and Suggestions for Authors

Dear authors,

The manuscript's topic is highly current and significant, focusing on the challenging issue of applying simulation conditions in paramedic training. This study attempted to contribute and provide empirical data on multiple aspects of using supraglottic airway devices for airway management in paramedic training.

However, major improvements are necessary for the manuscript to provide valid empirical data that will complement the existing literature on applying simulation conditions for airway management in paramedic training.

I want to make suggestions for improving the manuscript:

Title

The title should be concise and clearly indicate the participants in the study.

Abstract: Although the abstract is structured, significant revisions are needed:

  • The Aim is general; it is necessary to state the specific objectives of the study.
  • In the methods section, specify the study design, setting, participants and instrument used. At the same time, the description of the scenario should be shortened.

Keywords:

- Incomplete concerning the manuscript's content.

Introduction:
Inadequate literature review. Revisions are necessary for the following:

  • The introduction does not sufficiently address the existing literature on all aspects of the research presented in the manuscript and the subsequent parts. It is necessary to provide data on the application and importance of simulation in paramedic education. It would be beneficial to provide data on the country where the research was conducted, including the extent to which and in what form placement of supraglottic airways is studied during paramedic education, especially whether and what type of simulation is mandatory. Clarify whether simulation is the subject of the research or the research method. Please include this data.
  • Support the statements (lines 35-43; 47-52) with relevant references.
  • The Aim is general; it is necessary to state the specific objectives of the study.

Materials and Methods

The methodology partially allows for the replication of the study. However, it is challenging to follow the methodological framework of the study in this form, as certain segments are missing. Furthermore, it would be beneficial to provide data on whether the study adheres to the standards for reporting RCT trials (CONSORT) - Simulation-Based Research Extensions for the STROBE Statement (Reporting guidelines for healthcare simulation research: extensions to the CONSORT and STROBE statements). Also, I suggest you create subsections:

  • Study design and setting - Describe the setting, including locations and relevant dates, as well as the periods of recruitment, exposure, follow-up, and data collection.
  • Sample and data collection – specify the methods used for participant selection and the criteria for inclusion in the study. Understanding the entire research process would be much easier if a flowchart were provided, as the data presented in its current form is difficult to follow.
  • Study procedures – It is mostly correctly stated. Data are missing on: the theoretical concept of the scenario, who created the scenario, who selected the scenario, who were the participants and the criteria for selecting participants in the pilot study, whether the data of the participants in the pilot study were included in this manuscript, whether and what revisions of the initial scenario were made; who monitored the work of the participants in the study; whether and how often the SDA changes during the study.
  • Instruments – Concrete data on the instruments, method, and time of data collection are missing: Which instruments were used, how they were created (theoretical framework supported by references), the questions included, and the scoring criteria.
  • Data analysis
  • Ethical Consideration – Data for these exists in the text; separate them into separate subsections.

Results

The results presented in this form are difficult to understand. To facilitate comprehension, match the names of the tables with the accompanying text. Instead of 'Studied Device,' it would be more appropriate to use the name' Airway devices.' Tables must be formatted in accordance with the journal's template recommendations. It is also necessary to include all the data for statistical analysis, not just p-values; calculating effect sizes would also be important.

Discussion

The discussion is extensive; some parts could be moved to the introductory section. To achieve a logical structure after correcting the previous sections, they also need to be reviewed. Considering that more than two-thirds of the references are older than five years, it would be desirable to consider the study's results in light of more recent empirical data.

Conclusions

As a section, it is not mandatory, according to the instructions for authors. Still, considering the importance of the manuscript topic, I suggest that the authors create it as a separate section.

I hope you find my comments helpful

Author Response

Dear Reviewer 1,

Author's point-by-point response to reviewer's comments:

Comment: I want to make suggestions for improving the manuscript:

Title

The title should be concise and clearly indicate the participants in the study.

Author's reply: The tile has been shortened and now includes the study participants.

Abstract: Although the abstract is structured, significant revisions are needed:

  • The Aim is general; it is necessary to state the specific objectives of the study.

Author's reply: The aim has been changed as per reviewer’s suggestion.

  • In the methods section, specify the study design, setting, participants and instrument used. At the same time, the description of the scenario should be shortened.

Author's reply: The methods section has been changed as per reviewer’s suggestion.

Keywords:

- Incomplete concerning the manuscript's content.

Author's reply: The keywords have been changed as per reviewer’s suggestion.

Introduction:
Inadequate literature review. Revisions are necessary for the following:

  • The introduction does not sufficiently address the existing literature on all aspects of the research presented in the manuscript and the subsequent parts. It is necessary to provide data on the application and importance of simulation in paramedic education. It would be beneficial to provide data on the country where the research was conducted, including the extent to which and in what form placement of supraglottic airways is studied during paramedic education, especially whether and what type of simulation is mandatory. Clarify whether simulation is the subject of the research or the research method. Please include this data.

Author's reply: The introduction has been re-written and a paragraph on simulation in paramedic education was also added.

  • Support the statements (lines 35-43; 47-52) with relevant references.

Author's reply: Lines 35-43 and 47-52 have been supported with relevant references.

  • The Aim is general; it is necessary to state the specific objectives of the study.

Author's reply: The aim has been changed as per reviewer’s suggestion.

Materials and Methods

The methodology partially allows for the replication of the study. However, it is challenging to follow the methodological framework of the study in this form, as certain segments are missing. Furthermore, it would be beneficial to provide data on whether the study adheres to the standards for reporting RCT trials (CONSORT) - Simulation-Based Research Extensions for the STROBE Statement (Reporting guidelines for healthcare simulation research: extensions to the CONSORT and STROBE statements).

Author's reply: A CONSORT flowchart has been added and is now marked as Figure 1.

Comment: Also, I suggest you create subsections:

  • Study design and setting - Describe the setting, including locations and relevant dates, as well as the periods of recruitment, exposure, follow-up, and data collection.
  • Sample and data collection – specify the methods used for participant selection and the criteria for inclusion in the study. Understanding the entire research process would be much easier if a flowchart were provided, as the data presented in its current form is difficult to follow.
  • Study procedures – It is mostly correctly stated. Data are missing on: the theoretical concept of the scenario, who created the scenario, who selected the scenario, who were the participants and the criteria for selecting participants in the pilot study, whether the data of the participants in the pilot study were included in this manuscript, whether and what revisions of the initial scenario were made; who monitored the work of the participants in the study; whether and how often the SDA changes during the study.
  • Instruments – Concrete data on the instruments, method, and time of data collection are missing: Which instruments were used, how they were created (theoretical framework supported by references), the questions included, and the scoring criteria.
  • Data analysis
  • Ethical Consideration – Data for these exists in the text; separate them into separate subsections.

Author's reply: Suggested subsections have been created with adequate descriptions.

Results

The results presented in this form are difficult to understand. To facilitate comprehension, match the names of the tables with the accompanying text. Instead of 'Studied Device,' it would be more appropriate to use the name' Airway devices.' Tables must be formatted in accordance with the journal's template recommendations. It is also necessary to include all the data for statistical analysis, not just p-values; calculating effect sizes would also be important.

Author's reply: The Results section has been changed as per reviewer’s suggestion.

Discussion

The discussion is extensive; some parts could be moved to the introductory section. To achieve a logical structure after correcting the previous sections, they also need to be reviewed. Considering that more than two-thirds of the references are older than five years, it would be desirable to consider the study's results in light of more recent empirical data.

Author's reply: The Discussion section has been reviewed. More recent references have been added. Although one has to bear in mind that there is a lack of evidence in the current literature with regards to the topic, especially SADs in entrapped patients.

Conclusions

As a section, it is not mandatory, according to the instructions for authors. Still, considering the importance of the manuscript topic, I suggest that the authors create it as a separate section.

Author's reply: This section has been created as per reviewer’s suggestion.

Reviewer 2 Report

Comments and Suggestions for Authors

The authors examine a significant and insufficiently investigated aspect of pre-hospital airway management—specifically, the efficacy of supraglottic airway devices (SADs) in simulated entrapped trauma patients.  Nonetheless, despite the admirable goal, the work is plagued by significant methodological, interpretational, and editing deficiencies that compromise its scientific validity and applicability to clinical practice.  I highly advise substantial modifications prior to reevaluation, or rejection in its present state.

 The study design exhibits insufficient external validity.  A manikin model in a tilted vehicle configuration fails to accurately simulate the anatomical, physiological, or environmental problems present in actual trauma situations (e.g., secretions, changed anatomy, hemorrhage, or patient mobility).  This constrains the clinical relevance of the findings.
 The randomization process is inadequately articulated ("randomly assign a number...").  There is no reference to allocation concealment or measures to mitigate operator bias.  Moreover, the absence of blinding for the investigators or evaluators introduced a significant risk of performance and detection bias.
 3. Participants lacked prior experience with certain examined devices (e.g., SLIPA and AuraGain), which intrinsically disadvantages these devices for performance.  The 10-minute lecture and 30-minute familiarization session are inadequate to address this gap, particularly when operators are already acquainted with devices such as the I-gel and LMA Supreme.
 The text references statistical tests but fails to present the actual test statistics (e.g., t-values, z-values), confidence ranges, or power calculations in the results.  This undermines the trustworthiness of reported p-values.  Furthermore, the authors cite a "previous unpublished pilot study" as the foundation for the sample size, which is unassessable.
 The introduction and discussion parts exhibit a deficiency in synthesis and critical analysis.  Citations might occasionally be superfluous or excessively concentrated on the authors' previous research.  There is a paucity of comparisons with actual clinical research or registries.  The assertion of innovation is exaggerated, as analogous studies (e.g., Pap et al., Martin et al.) are already present.
 Only the duration of ventilation, self-reported ease of use, and success rate were examined.  No evaluation of airway seal integrity, aspiration risk, or complications—elements essential in trauma airway management—has been conducted.
 Several figures (e.g., flow diagrams, device photographs, vehicle configurations) lack scientific merit and are more illustrative than helpful.  Figures 2–6 are superfluous and ought to be relocated to supplementary material or eliminated entirely.
 The paper is replete with grammatical errors and clumsy phrasing (e.g., "the use of the LMA Supreme was associated...").  It necessitates comprehensive editing for linguistic style and clarity.
 The title is excessively lengthy and ought to be condensed for clarity and search optimization.
 Certain references (e.g., Brain 1991) are superfluous for substantiating the utilization of SAD in 2025.  The citation strategy appears obsolete in certain sections.

Comments on the Quality of English Language

The manuscript contains numerous grammatical errors and awkward constructions (e.g., "the use of the LMA Supreme was associated..."). It requires thorough editing for English style and clarity.

Author Response

Dear Reviewer 2,

Author's point-by-point response to reviewer's comments:

Comment: The authors examine a significant and insufficiently investigated aspect of pre-hospital airway management—specifically, the efficacy of supraglottic airway devices (SADs) in simulated entrapped trauma patients.  Nonetheless, despite the admirable goal, the work is plagued by significant methodological, interpretational, and editing deficiencies that compromise its scientific validity and applicability to clinical practice.  I highly advise substantial modifications prior to reevaluation, or rejection in its present state.

The study design exhibits insufficient external validity.  A manikin model in a tilted vehicle configuration fails to accurately simulate the anatomical, physiological, or environmental problems present in actual trauma situations (e.g., secretions, changed anatomy, hemorrhage, or patient mobility).  This constrains the clinical relevance of the findings.

Author's reply: This was a manikin study as it is not possible to conduct a study (mainly due to ethical concerns) in the country where the study was done (Poland). Furthermore, according to available and recognised research databases such as Pubmed, Medline or Web of Science/Knowledge no human studies exist with regards to this topic i.e. airway management in entrapped patients. We do agree that it is not possible to translate/extrapolate all the results of a manikin study to human population and this was pointed out in the study limitations paragraph. On the other hand a manikin study offers several advantages: 

  1. there is no patient recruitment and therefore no risk of severe adverse effects
  2. they offer stable/controlled experimental conditions, which may aid comparative studies
  3. are ethically-approved
  4. the extensive use of manikins for airway management research has led many international committees such as the (ERC) European Resuscitation Council to base their guidelines on bench studies

Comment: The randomization process is inadequately articulated ("randomly assign a number...").  There is no reference to allocation concealment or measures to mitigate operator bias.  Moreover, the absence of blinding for the investigators or evaluators introduced a significant risk of performance and detection bias.

Author's reply: The description of the randomization process has been improved for clarity and quality.

Comment: Participants lacked prior experience with certain examined devices (e.g., SLIPA and AuraGain), which intrinsically disadvantages these devices for performance.  The 10-minute lecture and 30-minute familiarization session are inadequate to address this gap, particularly when operators are already acquainted with devices such as the I-gel and LMA Supreme.

Author's reply: We agree that experience in use of the two above-mentioned airway devices can only be achieved by frequent use of them in routine day-to-day practice. At first, the times may look as insufficient but the aim was to allow paramedics to familiarize themselves with the new equipment. We set up both the lecture time as well as the practice time based on our previous experience when we evaluated various new airway devices both in pre-hospital as well as clinical settings. We agree that a considerable amount of time is required to master the technique. On the other hand, in the current literature there are studied available with minimal pre-assessment training e.g. Pius J, Noppens RR. Learning curve and performance in simulated difficult airway for the novel C-MAC® video-stylet and C-MAC® Macintosh video laryngoscope: A prospective randomized manikin trial. PLoS One 2020; 19: 15 e0242154. doi: 10.1371/journal.pone.0242154 or Mahli N, Md Zain J, Mahdi SNM, Chih Nie Y, Chian Yong L, Shokri AFA, Maaya M. The Performance of Flexible Tip Bougie™ in Intubating Simulated Difficult Airway Model. Front Med (Lausanne) 2021; 7; 8: 677626. doi: 10.3389/fmed.2021.677626.

Comment: The text references statistical tests but fails to present the actual test statistics (e.g., t-values, z-values), confidence ranges, or power calculations in the results.  This undermines the trustworthiness of reported p-values.  Furthermore, the authors cite a "previous unpublished pilot study" as the foundation for the sample size, which is unassessable.

Author's reply: The entire methods section has been reviewed and re-organised into subsections. This improves the clarity of the manuscript. The description of sample size calculations has re-written and other statistical values such as Cohen’s d have been added.

Comment: The introduction and discussion parts exhibit a deficiency in synthesis and critical analysis.  Citations might occasionally be superfluous or excessively concentrated on the authors' previous research. There is a paucity of comparisons with actual clinical research or registries.  

Author's reply: Both the Introduction as well as the Discussion have been reviewed and new references have been added to reflect the more recent studies. However, one has to bear in mind that it is extremely difficult to compare the results of our study to similar studies as there is only a small number of such studies available in the current literature.

Comment: The assertion of innovation is exaggerated, as analogous studies (e.g., Pap et al., Martin et al.) are already present.

Author's reply: We believe that our study is a novel one as no studies exist which would evaluate SADs in simulated conditions with a limited access to a patient who is entrapped in a tilted vehicle. Of note is fact that such a scenario i.e. an RTA resulting in a tilted vehicle with entrapped occupants is not uncommon in a real life situation. NB. Studies by Pap et al. and Martin et al. included other airway devices such as video laryngoscopes and the car in their studies was in a normal up right position which not always the case in real life RTAs.

Comment: Only the duration of ventilation, self-reported ease of use, and success rate were examined.  No evaluation of airway seal integrity, aspiration risk, or complications—elements essential in trauma airway management—has been conducted.

Author's reply: We agree with this comment and understand that evaluation of SADs is more complex and includes other features such as seal pressure. However, inclusion of all of these features would overload the study and distort the reader’s attention away from the main findings. This in turn could make the results difficult to conclude. That is why we chose some features for clarity purposes. We plan to further evaluate SADs focusing on device seal pressures, aspiration prevention, etc.

Comment: Several figures (e.g., flow diagrams, device photographs, vehicle configurations) lack scientific merit and are more illustrative than helpful.  Figures 2–6 are superfluous and ought to be relocated to supplementary material or eliminated entirely.

Author's reply: Thank you for this comment. While some figures seem to be unnecessary, we believe that they could be included to improve the clarity of the manuscript and to visualise the actual scenario and what was evaluated. Flow diagrams such as CONSORT enhance the value of a manuscript. Such an approach can also be found in similar studies which evaluated SADs in trauma patients.

Comment: The paper is replete with grammatical errors and clumsy phrasing (e.g., "the use of the LMA Supreme was associated...").  It necessitates comprehensive editing for linguistic style and clarity.

Author's reply: The entire manuscript has been re-read and spell-checked by a native English speaker.

Comment: The title is excessively lengthy and ought to be condensed for clarity and search optimization.

Author's reply: The tile has been shortened and now includes the study participants.

Comment: Certain references (e.g., Brain 1991) are superfluous for substantiating the utilization of SAD in 2025.  The citation strategy appears obsolete in certain sections.

Author's reply: We agree that some of the references seem to be outdated. However, they refer to certain cornerstones in airway management (such as Brain’s invention - the LMA) and therefore were included in the manuscript to highlight its importance in developing new airway devices. 

Comments on the Quality of English Language

The manuscript contains numerous grammatical errors and awkward constructions (e.g., "the use of the LMA Supreme was associated..."). It requires thorough editing for English style and clarity.

Author's reply: Thank you for your comment. The entire manuscript has been re-read and spell-checked by a native English speaker.

Reviewer 3 Report

Comments and Suggestions for Authors

This research is very interesting to reviewers. I have only minor comments in the method section for its scientific clarity.

Randomization Process: While participants selected a number corresponding to each device, it is unclear whether this method ensured true randomization or risked allocation bias. Please clarify how random sequence generation was ensured and whether any randomization software or protocol was used.

Blinding: Was there any attempt to blind the participants or assessors to device types or outcome measurements? Even in a manikin study, lack of blinding could introduce operator bias, especially in subjective assessments like ease-of-use.

Familiarization Period: Participants were unfamiliar with the SLIPA and AuraGain devices prior to the study. Although a 30-minute familiarization session was provided, this could still introduce a performance bias. Please discuss this limitation and whether device order was counterbalanced across participants to mitigate learning effects.

Data Recording Method: While time was measured using a mobile phone stopwatch, it would be beneficial to address any steps taken to standardize timing (e.g., who started/stopped the timer, was this done by the same person, etc.) to ensure consistency.

Statistical Analysis Tools: Analysis was conducted using Microsoft Excel. While acceptable for basic statistics, Excel has limitations for rigorous analysis. Consider confirming results with statistical software like SPSS, R, or STATA to ensure robustness, particularly for tests like the Kolmogorov-Smirnov.

Sample Size Justification: The justification of the sample size based on an assumed 90% success rate is appreciated, but additional details (e.g., effect size used in the power calculation) would increase transparency.

Author Response

Dear Reviewer 3,

Author's point-by-point response to reviewer's comments:

Comment: This research is very interesting to reviewers. I have only minor comments in the method section for its scientific clarity.

Randomization Process: While participants selected a number corresponding to each device, it is unclear whether this method ensured true randomization or risked allocation bias. Please clarify how random sequence generation was ensured and whether any randomization software or protocol was used.

Author's reply: The description of the randomization process has been improved for clarity and quality.

Comment: Blinding: Was there any attempt to blind the participants or assessors to device types or outcome measurements? Even in a manikin study, lack of blinding could introduce operator bias, especially in subjective assessments like ease-of-use.

Author's reply: The description of the blinding process has been improved and included in the Methods section.

Comment: Familiarization Period: Participants were unfamiliar with the SLIPA and AuraGain devices prior to the study. Although a 30-minute familiarization session was provided, this could still introduce a performance bias. Please discuss this limitation and whether device order was counterbalanced across participants to mitigate learning effects.

Author's reply: No previous experience with some of the studied devices can potentially be a source of bias and that is why it was discussed in a relevant paragraph of the manuscript.

Comment: Data Recording Method: While time was measured using a mobile phone stopwatch, it would be beneficial to address any steps taken to standardize timing (e.g., who started/stopped the timer, was this done by the same person, etc.) to ensure consistency.

Author's reply: Thank you for pointing this out. We added more detailed description of the recording method.

Comment: Statistical Analysis Tools: Analysis was conducted using Microsoft Excel. While acceptable for basic statistics, Excel has limitations for rigorous analysis. Consider confirming results with statistical software like SPSS, R, or STATA to ensure robustness, particularly for tests like the Kolmogorov-Smirnov.

Author's reply: Thank you for this comment. We used different software for statistical analysis e.g. Excel, SPSS and Statistica. A brief description has been added in the Data Analysis subsection of the Methods section.

Comment: Sample Size Justification: The justification of the sample size based on an assumed 90% success rate is appreciated, but additional details (e.g., effect size used in the power calculation) would increase transparency.

Author's reply: Effect size was calculated and Cohen’s d included in the text.

Round 2

Reviewer 2 Report

Comments and Suggestions for Authors

article can be accepted in current form